# Long-Term Complications of Radioligand Therapy with Lutetium-177 and Yttrium-90 in Patients with Neuroendocrine Neoplasms

**DOI:** 10.3390/nu15010185

**Published:** 2022-12-30

**Authors:** Marek Saracyn, Adam Daniel Durma, Barbara Bober, Maciej Kołodziej, Arkadiusz Lubas, Waldemar Kapusta, Stanisław Niemczyk, Grzegorz Kamiński

**Affiliations:** 1Department of Endocrinology and Radioisotope Therapy, Military Institute of Medicine—National Research Institute, 04-141 Warsaw, Poland; 2Department of Internal Diseases, Nephrology and Dialysis, Military Institute of Medicine—National Research Institute, 04-141 Warsaw, Poland

**Keywords:** RLT, PRRT, myelotoxicity, nephrotoxicity, renal, hematological, chronic complications, IL-18, KIM-1, 177-Lu, 90-Y

## Abstract

Background: Neuroendocrine neoplasms are a group of tumors deriving from neural crest. They can be located in every tissue, but most commonly in gastrointestinal tract. Targeted therapy with use of radionuclides is an available and acceptable way of treatment, but its long-term safety is still to be determined, especially with sensitive methods. Methods: Study was performed on a group of 42 patients. They underwent full cycle (4 courses; 8–12 weekly intervals) of radioligand therapy with [^177^Lu]Lu-DOTATATE alone or tandem therapy with [^177^Lu]Lu-DOTATATE+[^90^Y]Y-DOTATATE. Late and long-term marrow and renal complications were assessed. Analysis focused on comparing data before first, fourth, and one year after the last course of RLT. Results: Study showed decreasing of all blood parameters in long-term observation, especially in lymphocytes line. Type of radioisotope, other diseases, primary tumor location, BMI, gender or age did not affect results. The only factor that had influence on hemoglobin and erythrocytes was decreased renal filtration. In long-term observation almost 10% decrease of renal filtration was observed. Type of isotope, gender, age, BMI did not affect these results. Moreover, reduction of urine IL-18, KIM-1, and albumin concentration has been observed. Conclusions: Though low-grade complications of radioligand therapy are possible, it stay a safe method of NEN treatment where benefits outweigh the risk.

## 1. Introduction

Neuroendocrine neoplasms (NENs) are a group of neoplasm arising from embryonic neural crest, both neuroectoderm and endoderm [1]. They can be found in almost every part of human body, but mainly in gastrointestinal system. The most common location of NENs is small intestine [2]. Majority of NENs are non-functioning tumors, with slow growth, and no clinical symptoms in early stages. They are recognized accidentally, the most often when the liver metastases appear [2,3].

Radioligand therapy (RLT), previously defined as peptide receptor radionuclide therapy (PRRT), is usually second line of treatment, introduced after inefficiency or no possibility of primary treatment, and with disease progression [2]. RLT is typically recommended in G1 and G2 grading of NEN. There is a possibility to use this treatment in NENs G3, but only when somatostatin receptors expression is confirmed in somatostatin receptor imaging (SRI) using ^99m^Tc scintigraphy or ^68^Ga-PET [4]. Presently the most common radioisotopes used for treatment are beta-emitters: ^177^Lutetium, and ^90^Yttrium. ^177^Lutetium characterizes energy (Emax) of 0.497 MeV, range of 2 mm, and half-life of 6.647 days, ^90^Yttrium emits β-radiation with energy (Emax) of 2.27 MeV, range of 11 mm, and half-life of 2.67 days. Due to shorter range and lower energy ^177^Lutetium is theoretically less myelo- and nephrotoxic. There are also attempts to administered both radioisotopes simultaneously as a tandem therapy [5]. Most common reasons of therapy withdrawal are myelo-, hepato- or nephrotoxicity [6].

## 2. Materials and Methods

### 2.1. Patients and Protocol

From 2017 to 2020—42 patients who were qualified to RLT, and agreed to take part in the study, were hospitalized in Endocrinology and Radioisotope Therapy Department of Military Institute of Medicine. Informed consent was obtained from all patients involved in the study. The study was conducted to the guidelines of Helsinki Declaration, and approved by local Bioethical Committee (52/WIM/2017). 31 of patients received intravenously 7.4 GBq (200 mCi) of [^177^Lu]Lu-DOTATATE, and 11 received tandem therapy with [^90^Y]Y-DOTATATE + [^177^Lu]Lu-DOTATATE (in 1.85 GBq/1.85 GBq, i.e., 50 mCi/50 mCi ratio). 36 patients underwent a full cycle of 4 RLT courses, with 8–14 weeks interval between. Long term observation was made in 25 patients, mainly due to SARS-CoV2 pandemic. During the treatment, nephroprotection (intravenous amino acids) was given directly before, and a day after each RLT course. During intervals, long-lasting somatostatin analogues: lanreodie 120 mg, or octreotide 30 mg (60% vs. 40%, respectively) were administrated every 4 weeks. Laboratory parameters were assessed before every course, and a year after last (IV) course. During the study we focused on parameters that could have potential predictive value in indicating bone marrow and renal complications. Characteristic of the study group is presented in Table 1.

### 2.2. Inclusion and Exclusion Criteria

Qualification to RLT was made due to commonly accepted international recommendation, and final decision was taken by Tumor Board Meeting. All patients had histologically confirmed NEN, good expression of somatostatin receptors in somatostatin receptor imaging (SRI) performed 12 weeks before treatment, and assessed stage of the disease in morphological examination (computed tomography or magnetic resonance imaging). Directly before RLT administration clinical state, total blood count, renal and liver parameters were also checked. Detailed inclusion criteria were as follows:Well- and moderately-differentiated unresectable metastatic progressive neuroendocrine neoplasm (defined as Ki-67 < 20%, progression according to the RECIST 1.1 (Response Evaluation Criteria In Solid Tumors) criteria, over the previous 12 months);Good expression of somatostatin receptors in qualifying somatostatin receptor scintigraphy (SRS) (SPECT/CT) (radiotracer uptake in the majority of the lesions higher than in normal liver (Krenning scale 3)) or in Gallium-68-PET/CT (SUVmax in the majority of the lesions higher than SUVmax in normal liver);No possibilities of surgical treatment;Chronic treatment with long-acting somatostatin analogues. Exclusion criteria was lack of consent, pregnancy or lactation, Karnofsk’y scale <60, WHO/ECOG 3 or 4, no tracer uptake in SRI, myelosuppression (understood as hemoglobin <8 g/L, or platelets <80.000/µL, or leukocytes <2000/µL, or lymphocytes <500/µL, or neutrophils <1000/µL), renal disfunction (eGFR <30 mL/min, or serum Creatinine >1.8 mg/dL) and liver diseases (ALT 3× over upper limit) [2].

### 2.3. Laboratory Evaluation

To examine potential distant effect on bone marrow and kidneys, we compared results before course I, course IV, and a year after the last course of RLT. For the purposes of this study the definition of late evaluations/complications means differences between course I and IV, follow-up—comparison between course IV and a year after treatment, and long-term observation means differences between course I and a year after treatment.

Statistical analysis was performed with the IBM SPSS Statistics package (Version 25.0., Armonk, NY, USA: IBM Corp. (Released 2021)). It was used to perform the analyses of basic descriptive statistics with the Shapiro–Wilk test and two-way mixed analysis of variance. Differences between dependent variables were analyzed with the use of the appropriate T-test or the Wilcoxon test. Differences between groups were analyzed with the appropriate T-test or the Mann–Whitney U test. Correlations between variables were analyzed using Pearson’s or Spearman’s test. A *p* value <0.05 was assumed as the level of significance. All data are expressed as means and standard deviation.

## 3. Results

### 3.1. Course I vs. Course IV—The Late Complications

Comparing results of course I and course IV, we have shown significant decrease of all blood parameters, except reticulocytes. The biggest differences were observed in lymphocytes number (−0.83 10^3^/µL; −47.70%; *p* < 0.001) (Table 2). Type of radioisotope, age, gender, body mass index (BMI), other chronic diseases, and location of primary tumor did not significantly affected the results.

Analysis of renal parameters showed significant decrease of Kidney Injury Molecule 1 (KIM-1), interleukin 18 (IL-18) and albumin urine concentration (*p* = 0.003; *p* < 0.001; *p* = 0.011, respectively) in late observation. IL-18 concentration decrease was more significant in patients with diabetes (*p* = 0.003), and those treated with [^177^Lu]Lu-DOTATATE/[^90^Y]Y (*p* = 0.013). KIM-1 was also more decreased in patients treated with tandem therapy (*p* = 0.012). In late observation, no GFR change was noticed. Detailed renal parameters of late evaluation are presented in Table 3.

### 3.2. Course IV vs. One Year after Treatment—The Follow-Up

Comparing blood count parameters before course IV, and one year after the last one, we noticed improvement of all blood parameters, but statistical importance was only observed for lymphocytes (Δ = 0.29×10^3^/µL; 33.33%; *p* = 0.002) (Table 4). Again, radioisotope, age, gender, body mass index (BMI), other chronic diseases, and location of primary tumor did not significantly affected results.

Examination of renal parameters showed not significant decrease of eGFR (*p* < 0.06) in follow-up. No other measured factors had significant influence on the results. Detailed renal results of the evaluated period are presented in Table 5.

In the follow-up we did not notice statistical differences in lymphocytes or GFR between groups of patients treated with ^177^Lu alone or tandem therapy.

### 3.3. Course I vs. One Year after Treatment—The Long-Term Evaluation

In long-term observation, total blood count showed significant decrease in number of leukocytes (*p* = 0.002), neutrophils (*p* = 0.025), lymphocytes (*p* = 0.002), erythrocytes (*p* < 0.001) and concentration of hemoglobin (*p* = 0.009). Number of reticulocytes did not change (Table 6). There was also correlation between low eGFR (<60 mL/min/1.73 m^2^) and higher decrease in mean erythrocytes number (*p* = 0.017), and mean hemoglobin concentration (*p* = 0.034). Previous chemotherapy also positively correlated with higher decrease in erythrocytes number (*p* = 0.009).

Long-term observation showed significant decrease of GFR (*p* = 0.009), and increase of serum creatinine concentration (*p* = 0.036). Type of radioisotope, age, gender, BMI, other chronic diseases (diabetes, hypertension), previous chemotherapy, did not significantly affected the results.

The only factor affecting GFR decrease was extrapancreatic localization of the tumor (*p* = 0.031). There was also significant decrease of urine IL-18 concentration (*p* = 0.084). Mean KIM-1 urine concentration also decreased, but the results were statistically unsignificant (Table 7).

In long-term observation we did not notice statistical differences in blood or renal parameters between group of patients treated with ^177^Lu alone or tandem therapy.

### 3.4. Adverse Events Analysis

Adverse events (AE) were assessed with use of National Cancer Institute (NCI) Common Terminology Criteria for Adverse Events (CTCAE version 5.0). In our study no G4 and G5 adverse events were observed. All adverse events noted is listed below (Table 8).

## 4. Discussion

Our study showed a deterioration of marrow function during the RLT and partial improvement in the follow-up. Nevertheless, one year after the last course of RLT comparing to pre-treatment data, all blood parameters remained reduced. The greatest decrease was observed in lymphocytes line. Type of radioisotope, other diseases, primary tumor location, BMI, gender or age did not affected the results. The only factor, that had significant influence on red cell line parameters was GFR (<60 mL/min/1.73 m^2^). Probably, it is because of longer circulation of radioisotope in the blood, and initially lowered hematopoiesis in this subgroup of patients. During the study hematological G4 and G5 adverse events were not observed. The highest percentage of G3 adverse events was registered in lymphocytes number before course IV but number of AEs decreased significantly the year after the last course.

We design our study to find a possible marker of nephrotoxicity after RLT. We chose urine Kidney Injury Molecule 1(KIM-1) which is a sensitive, and known marker of acute kidney injury. It is a superficial antigen located on renal tubule cells which can be used to assess its injury of different origin. Urine interleukin 18 (IL-18) is a marker of inflammatory process, mainly in the renal interstitium. So, it was used as another marker to assess kidney injury, in deeper renal tissues. Albuminuria is sensitive marker of injury of the renal filtration barrier. Its concentration is also related to endothelial damage.

In the long-term observation almost 10% decrease of glomerular filtration was observed. However, again, radioisotope, gender, age, other diseases and BMI did not affected results. Only factor that affected GFR was extrapancreatic tumor location. Surprisingly, reduced albuminuria and urine IL-18 and KIM-1 concentration were observed in the follow-up and the long-time observation, though in the last case in statistically unsignificant manner. Deeper decrease of urine IL-18 concentration was observed in patients treated with tandem therapy.

Alike results regards to safety issues were obtained in some previous trials. Sitani et. all were analyzing retrospectively a group of 468 patients with metastatic or advanced NEN [7]. All patients underwent at least 2 cycles of RLT with use of 5.5 to 7.4 GBq ^177^Lu-DOTATATE administrated in 10–12 week intervals. Patients were observed for 4 to 96 months after treatment (M = 46 months). Results showed hematological toxicity of Grade 1 in 1.7%, Grade 2 in 0.2%, and Grade 3 in 0.2% patients. Nephrotoxicity of Grade 1, Grade 2, Grade 3, and Grade 4 were seen in 3.5%, 0.6%, 0.4% and 0.2% patients, respectively. On the other hand Sarit et al., analyzed 78 patients who underwent at least one of four cycles of ^177^Lu-DOTATATE (7.4 GBq per dose), separated by 8-week intervals. They noticed G1-G2 adverse events in 60.3% patients, with most common one—G2 leukopenia, which was found in 33.3% patients. From 55 patients, that underwent a full cycle of treatment G2 leukopenia was observed in 23.6%. Grade 3 or 4 adverse events were observed in 32.1% patients. The most common was decrease of erythrocytes and leukocytes—observed in 12.8% patients. No chronic kidney injury was assessed in this study [8]. The difference in observed adverse events percentage may arise from fact, that Sarit et al., were analyzing acute and chronic complications combined in all timeline of the observation. Nevertheless, there are some cases of serious adverse events (SAEs) during RLT in the literature, but can be considered as very rare [9].

Bodei at al., on the group of 807 patients were comparing nephrotoxicity of different types of radioisotopes. They confirmed that treatment with ^177^Lu compared to ^90^Y and ^177^Lu/^90^Y  is less nephrotoxic. Renal AEs was observed in 13.4%; 33.6%; 25.5%, respectively (*p* < 0.001) [10]. This study confirms theoretical features of radioisotopes ensuring safer profile of ^177^Lu administrated alone. In our study we also observed, that use of tandem ^177^Lu/^90^Y (1.85/1.85 GBq) treatment compared to ^177^Lu alone (7.4 GBq) can give higher rate of long-term complications in total blood count and renal parameters, however some of the results were only at statistical trend level. Previous studies also pointed a lower rate of complications when only ^177^Lu was administrated [11,12]. Statistical differences may arise from different radioisotopes activities administered, treatment and nephroprotection protocols, or observation periods.

Bergsma at al. indicate that lowered GFR can be associated with higher number of myelotoxicity, due to extended time of radioisotope presence in bloodstream [13]. In the group of 323 patients they observed average annual GFR decrease of 3.4% after RLT with use of ^177^Lu. They observed as well that hypertension, diabetes, high cumulative activity of radioisotope, and initial CTCAE grade had no significant effect on renal function in long-time observation. These conclusions were confirmed by us. The annual glomerular filtration decrease in our study was 9.2% in the whole group, but 6.7% and 12.9% in the ^177^Lu and the tandem group, respectively. In the long-term observation, after 18 months since RLT start, the difference was even greater, i.e., 6.3% and 17.3% for 177Lu and tandem group, respectively, although both analyses did not reached statistical significance. It is worth noting, because 9.2% annual decrease of GFR is much higher than one in normal after 40 year population, which includes in 1–2%.

Interesting study was made by Scalorbi et. all, where authors tried to find predictive factors of myelo-, nephro-, and hepatotoxicity by using a Firth’s logistic regression with intercept correction (FLIC) model [14]. On the cohort of 87 patients treated with ^177^Lu-Oxodotreotide, (7.4 GBq iv per administration, with 8 ± 2 weeks interval) the subgroup of 67 patients—36 females and 31 males, with mean age of 63 was retrospectively analyzed. In those patients at least one G1-G2 AE were noted, while G3-G5 were casuistic. No renal G3-G4 adverse events were reported. The most observed compilation was GFR decrease (75.9%), anemia (68.6%) thrombocytopenia (47.8%) and leukopenia (44.8%). In the study, gender was a predictor of anemia, leukopenia, thrombocytopenia, and creatinine increase. Previous chemotherapy was not a predictive factor of AE onset, which was confirmed by other authors [13,15]. In our study, we observed that chemotherapy was the only factor that had an influence on decrease in erythrocytes number. These results may arise from different type of chemotherapy, which was used in both trials. Scalorbi et al., also noted there is an association between splenectomy and the risk of hematological complications, which can suggest that spleen removal can be a protective factor of this type of toxicity [14]. The limitation of that study is relatively short time of observation, which was 30 weeks from the beginning of the therapy.

Just like Theiler at al. in the group of 116 patients that was separated in elderly cohort (n = 48; age 81.7 ± 1,5), and younger cohort (n = 68; age 67.6 ± 1,7), we also confirmed that older age of patients is not connected with higher number of adverse events in long-term observation. This conclusions confirms the fact that RLT can be administrated in all age groups [16].

In our study we tried to prove utility of IL-18, or KIM-1 concentrations as markers of nephrotoxicity during and after RLT. We suspected, that those markers, which concentration increases in toxic or ischemic kidney injury, will be also elevated after RLT [17,18,19,20,21]. Results were completely different, as we observed decrease concentration of those parameters in late and long-term observation. It can be linked to potential immunosuppressive influence of RLT. Another explanation is that RLT can inhibit synthesis of KIM-1 and IL-18. Bogdándi et al., in experiment on C57BL/6 mice irradiated with different doses of gamma radiation noticed decrease in expression of the T-helper 1 and 2 (Th1, Th2) type cytokines after low doses (and increase after high doses). Interleukin 6 (IL-6) reacted earlier and IL-10 later. In the study, external source of gamma radiation was used, nevertheless we have to remember about partial spectrum of gamma radiation emitted by ^177^Lu and ^90^Y, and its possible local effect in kidneys [22]. Potential immunomodulate effect of gamma radiation on tissues is also described in human [23,24,25,26]. There is no clear evidence, that gamma or beta radiation can cause similar response in patients treated with RLT, so to confirm this hypothesis, and phenomenon of decreased KIM-1 and IL-19 after RLT, further studies are necessary.

Radioligand therapy, is becoming more preferrable line of treatment, due to results of many studies, especially NETTER-1, which proved 11/7 month difference in median overall survival with ^177^Lu treatment (8 cycles of 200 mCi; 8 weeks interval) plus 30 mg octreotide im. monthly versus octreotide alone (60 mg octreotide im. monthly). Despite not reaching statistical significancy in improving median overall survival it showed good safety profile of the treatment and confirmed its well-tolerable profile [27]. Presently RLT is recommended by majority of endocrine/oncological societies as a second line of treatment, when progression during somatostatin analogues is noted [2]. Both lanreotide and octreotide confirmed its efficiency versus placebo in NENs treatment, but long-term effectiveness of this drugs is questionable [28,29,30]. Other used ways of treatment are inhibitors of mammalian target of rapamycin (mTORi) like everolimus, ultrasound-guided-radioablation of tumor, or chemotherapy. Everolimus compared with placebo is connected with prolonged progression-free survival in patients with NENs, but has many side effects [31,32], and Bison et al., in their study proved that combination of RLT with mTOR was less effective than RLT alone [33]. Tumor radioablation is limited by tumor location, size, and primary staging. It is also expensive and demand experienced team [34]. Chemotherapy is usually last line of treatment because of its adverse events profile and effectiveness related to primary tumor location [2,35,36,37,38].

## 5. Study Limitations

The study was conducted in a relatively low number of patients, mostly because of the low incidence of neuroendocrine neoplasms, and a fact, that not all patients hospitalized in clinic agreed to take part in it. Another limitation was lower number of patients that took part in long-term control, but it was mainly due to COVID-19 pandemic. However, the study was not an epidemiological one, and was aimed at assessing the long-term complications of radioisotope treatment in a selected group of patients.

## 6. Study Strengths

The study was a prospective one, with aim to assess long-term complication and measure factors, that could be useful as markers of bone marrow and kidney injury. No previous studies were made in this context to estimate KIM-1 and IL-18 utility.

## 7. Conclusions

The radioligand therapy in patients with neuroendocrine neoplasms caused long-term hematological complications, especially noticed in lymphocytes line. Only erythrocytes decrease was correlated with decreased GFR and previous chemotherapy. Type of RLT, gender, age, BMI, primary tumor location, others diseases did not influenced the results.

In long-term observation RLT caused a significant almost 10% decrease of GFR, regardless of, age, gender, BMI, and other diseases. Deeper decrease of GFR was observed in patients treated with tandem therapy, but in statistically unsignificant manner.

KIM-1 and IL-18 concentrations did not proved its value as markers of kidney injury after radioligand therapy.

Complications of the treatment, assessed according to the international classification of adverse events, were mainly first and second degree, with exceptions of lymphopenia and deterioration of glomerular filtration, where single causes of third degree were noted.

The study confirmed that RLT is a safe method of NEN treatment, without high risk of serious adverse events (SAEs).

## Figures and Tables

**Table 1 nutrients-15-00185-t001:** Characteristic of the study group.

Number of Patients	N	**42**
Age	mean	58.1 ± 13.1
	range	23–76
Gender	females	19 (45.2%)
	males	23 (54.8%)
BMI	mean	24.9 ± 5.2
	range	16.4–41.3
	<18.5	3 (7.1%)
	28.5–24.9	21 (50%)
	24.9–29.9	12 (28.6%)
	≥30.0	6 (14.3%)
Primary NEN location	pancreas	15 (35.6%)
	small intestine	13 (30.9%)
	large intestine	5 (12%)
	other (lungs, ovaries, stomach)	5 (12%)
	unknown	4 (9.5%)
NEN Staging	G1	20 (48%)
	G2	22 (52%)

BMI—Body Mass Index, NEN—Neuroendocrine neoplasm; N—number of patients.

**Table 2 nutrients-15-00185-t002:** Blood parameters before course I and IV of RLT.

TBC	U	Course I (n = 36)		Course IV (n = 36)		Δ	Δ %	*p*
		M	SD	M	SD			
WBC	[10^3^/µL]	7.15	1.83	4.81	1.93	−2.34	−32.73	<0.001
NEU	[10^3^/µL]	4.67	1.56	3.29	1.65	−1.38	−29.55	<0.001
LYM	[10^3^/µL]	1.74	0.83	0.91	0.44	−0.83	−47.70	<0.001
RBC	[10^6^/µL]	4.52	0.59	3.88	0.67	−0.64	−14.16	<0.001
HGB	[g/dL]	13.59	1.74	12.27	1.62	−1.32	−9.71	<0.001
PLT	[10^3^/µL]	250.44	112.14	183.64	52.96	−66.8	−26.67	0.027
RET	[%]	1.51	0.44	1.72	0.77	0.21	13.91	0.126

TBC—total blood count; U—units; M—mean value; SD—standard deviation; Δ—change of values; Δ %—change of values (%); WBC—white blood cells; NEU—neutrophils; LYM—lymphocytes; RBC—erythrocytes; HGB—hemoglobin; PLT—platelets; RET—reticulocytes.

**Table 3 nutrients-15-00185-t003:** Renal parameters before course I and IV.

Renal Parameters	U	Course I (n = 33)		Course IV (n = 33)		*p*
		M	SD	M	SD	
Cr_s_	[mg/dL]	0.91	0.28	0.92	0.31	0.388
GFR CKD-EPI cr	[mL/min/1.73 m²]	86.57	26.33	84.74	24	0.93
Urine albumins	[mg/mL]	7.84	19.14	2.89	5.79	0.011
Urine KIM-1	[pg/dL]	1851.14	1343.97	1416.09	1393.1	0.003
Urine IL-18	[pg/dL]	167.34	126.48	47.94	58.1	<0.001

U—units; M—mean value; SD—standard deviation; GFR CKD-EPI—estimated glomerular filtration rate according to Chronic Kidney Disease Epidemiology Collaboration; Cr_s_—serum creatinine;, KIM-1—kidney injury molecule 1; IL-18—interleukin 18.

**Table 4 nutrients-15-00185-t004:** Blood parameters before course IV and the year after course IV.

TBC	U	Course IV (n = 36)		One Year after Treatment (n = 36)		Δ	Δ %	*p*
		M	SD	M	SD			
WBC	[10^3^/µL]	4.57	2.21	5.26	1.95	0.69	15.1	0.426
NEU	[10^3^/µL]	3.12	1.8	3.43	1.57	0.31	9.94	0.957
LYM	[10^3^/µL]	0.87	0.49	1.16	0.64	0.29	33.33	0.002
RBC	[10^6^/µL]	3.88	0.76	3.89	0.82	0.01	0.26	0.786
HGB	[g/dL]	12.22	1.74	12.28	1.8	0.06	0.49	0.936
PLT	[10^3^/µL]	175.47	53.85	189.95	93.28	14.48	8.25	0.807
RET	[%]	1.9	0.77	1.76	0.67	−0.14	−7.37	0.386

TBC—total blood count; U—units; M- mean value; SD—standard deviation; Δ—change of values; Δ %—change of values (%); WBC—white blood cells; NEU—neutrophils; LYM—lymphocytes; RBC—erythrocytes; HGB—hemoglobin; PLT—platelets; RET—reticulocytes.

**Table 5 nutrients-15-00185-t005:** Renal parameters before course IV and the year after course IV.

TBC	U	Course IV (n = 19)		One Year after Treatment (n = 19)		*p*
		M	SD	M	SD	
Creatinine	[mg/dL]	0.95	0.35	1.04	0.42	0.155
eGFR CKD-EPI cr	[mL/min/1.73 m²]	87.74	30.82	80.95	32.25	0.06
Urine albumins	[mg/mL]	1.96	2.83	4.8	8.46	0.147
urine KIM-1	[pg/mL]	1839	2443.06	934.65	615.98	0.564
urine IL-18	[pg/mL]	65.95	73.18	37.26	18.16	0.856

U—units; M—mean value; SD—standard deviation; eGFR CKD-EPI—estimated glomerular filtration rate Chronic Kidney Disease Epidemiology Collaboration; Cr_s_ serum creatinine; KIM-1—kidney injury molecule 1; IL-18—interleukin 18.

**Table 6 nutrients-15-00185-t006:** Blood parameters before course I and the year after course IV.

TBC	U	Course I (n = 19)		One Year after Course IV (n = 19)		Δ	Δ %	*p*
		M	SD	M	SD			
WBC	[10^3^/µL]	6.59	1.50	5.26	1.95	−1.33	−20.18	0.002
NEU	[10^3^/µL]	4.29	1.21	3.43	1.57	−0.86	−20.05	0.025
LYM	[10^3^/µL]	1.64	0.96	3.43	0.64	−0.48	−29.27	0.002
RBC	[10^6^/µL]	4.61	0.58	3.89	0.82	−0.72	−15.62	<0.001
HGB	[g/dL]	13.62	1.45	12.28	1.80	−1.34	−9.84	0.009
PLT	[10^3^/µL]	262.79	150.68	189.95	93.28	−72.84	−27.72	0.317
RET	[%]	1.60	0.41	1.76	0.67	0.16	10.00	0.396

TBC—total blood count; U—units; M—mean value; SD—standard deviation; Δ—change of values; Δ %—change of values (%); WBC—white blood cells; NEU—neutrophils; LYM—lymphocytes; RBC—erythrocytes; HGB—hemoglobin; PLT—platelets; RET—reticulocytes.

**Table 7 nutrients-15-00185-t007:** Renal parameters in long-term observation.

	U	Course I (n = 19)		One Year after Treatment (n = 19)		*p*
		M	SD	M	SD	
Creatinine	[mg/dL]	0.91	0.27	1.04	0.42	0.036
eGFR CKD-EPI cr	[mL/min/1.73 m²]	88.11	26.47	80.95	32.25	0.009
Urine albumins	[mg/mL]	2.97	4.37	4.8	8.46	0.314
urine KIM-1	[pg/mL]	1799.83	1426.41	934.65	615.98	0.155
urine IL-18	[pg/mL]	183.80	145.70	37.26	18.16	0.084

U—units; M—mean value; SD—standard deviation; eGFR CKD-EPI—estimated glomerular filtration rate Chronic Kidney Disease Epidemiology Collaboration; Cr creatinine; KIM-1—kidney injury molecule 1; IL-18—interleukin 18.

**Table 8 nutrients-15-00185-t008:** Adverse events summary.

	Course I (n = 42) (%)		Course IV (n = 35) (%)			One Year after Treatment (n = 19) (%)		
Stage of AE	G1	G2	G1	G2	G3	G1	G2	G3
Leukopenia	1 (2.3%)	0 (0%)	5 (14.3%)	6 (17.1%)	0 (0%)	0 (0%)	4 (21%)	0 (0%)
Neutropenia	0(0%)	0 (0%)	5 (14.3%)	6 (17.1%)	0 (0%)	1 (5.2%)	3 (15.8%)	0 (0%)
Lymphopenia	2 (4.6%)	2 (4.6%)	4 (11.4%)	13 (37.1%)	5 (14.3%)	1 (5.2%)	7 (36.8%)	1 (5.2%)
GFR decrease	17 (40%)	6 (14.3%)	13 (37.1%)	5 (14.3%)	0 (0%)	9 (47.4%)	0 (0%)	3 (15.8%)

## Data Availability

Data other than published in manuscript is partially unavailable due to privacy or ethical restrictions.

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
