# Peer review of "Long-Term Complications of Radioligand Therapy with Lutetium-177 and Yttrium-90 in Patients with Neuroendocrine Neoplasms"

_nutrients, 2022, doi:10.3390/nu15010185_

Round 1

Reviewer 1 Report

The article entitled “Long-term Complications of Radioligand Therapy with Lutetium-177 and Yttrium-90 in Patients with Neuroendocrine Neoplasms” by Saracyn et al shows that although low-grade complications of are possible, radioligand therapy stays a safe method of NEN treatment where benefits outweigh the risk.

Reduction of urine IL-18, KIM-1, and albumin concentration following radioligand therapy is important new information, providing clinical parameters to monitor for efficacy of therapy.

The paper is excellent as submitted. The manuscript addresses an important clinical problem, the study is well executed, the statistics robust, and the presentation clear.

I have just a few suggestions:

1). Please consider adding a few lines to the discussion making comparison to other therapies, and why radioligand therapy is superior. Lanreotide has questionable efficacy. Everolimus has many side effects. Radio ablation is invasive, expensive, and application is limited by tumor location and size.

2). Why KIM-1, IL-18, and albumin? We know, but some readers may not.

Author Response

Response to Reviewer 1 Comments

1). Please consider adding a few lines to the discussion making comparison to other therapies, and why radioligand therapy is superior. Lanreotide has questionable efficacy. Everolimus has many side effects. Radio ablation is invasive, expensive, and application is limited by tumor location and size.

Kindly appreciate al suggestions. Of course, we added few lines of comparsion, at the end of discussion

2). Why KIM-1, IL-18, and albumin? We know, but some readers may not.

As well, we added explanation in text why this parameters were chosen.

All changes marked red in manuscript

Reviewer 2 Report

A comparison between clinical effectivness of  the treatment and  late complications could increase the impact of the paper. 

Author Response

Response to Reviewer 1 Comments

A comparison between clinical effectivness of  the treatment and  late complications could increase the impact of the paper. 

In our study we did not focused on assessment of effectiveness of the treatment and its comparing with complications in treated patients. Suggestion is extremally useful, and we certainly take it under consideration in designing future studies.